# Short-term multicomponent exercise training improves executive function in postmenopausal women

Dani Rahmat Ramadhana[1], Rizki Prayuda Putra[1], Michelle Abigail Sibarani[1], Sulistiawati Sulistiawati[2], Dewi Ratna Sari[3], Purwo Sri Rejeki[1,4,5], Lilik Herawati[1,4,5], Raden Argarini[1,4,5]*

1 Faculty of Medicine, Master Degree on Sport, Exercise and Health Sciences, Airlangga University, Surabaya, Indonesia, 2 Faculty of Medicine, Department of Public Health and Preventive Medicine, Airlangga University, Surabaya, Indonesia, 3 Faculty of Medicine, Department of Anatomy, Histology and Pharmacology, Airlangga University, Surabaya, Indonesia, 4 Faculty of Medicine, Undergraduate Medical Programme, Airlangga University, Surabaya, Indonesia, 5 Faculty of Medicine, Department of Medical Physiology and Biochemistry, Airlangga University, Surabaya, Indonesia

* raden-a@fk.unair.ac.id

**Data Availability Statement:** All relevant data are within the manuscript and its Supporting Information files.

## Abstract

Declined cognitive function is commonly complained during the menopausal transition and continues afterward. Combining different exercises potentially leads to greater improvements in cognitive function, however, evidence of the benefits that accrue with multicomponent exercise training, specifically for postmenopausal women is limited. Therefore, this study aimed to investigate the effects of short-term multicomponent exercise training programs on executive function in postmenopausal women. Thirty women (59.8 ± 5.2 years), who were at least 12 months post menopause were allocated into a control (CON) group and an exercise (EX) training group. The EX group underwent a 2-week (five times/week) multicomponent exercise program comprising aerobic, strength, flexibility, and balance exercises for 40–60 min. Executive function was assessed by using the Stroop test and global cognitive function was assessed using the Mini-Mental State Examination (MMSE) at baseline (pre) and after 2 weeks (post) of exercise. The EX group showed improved performance in the Stroop test, with faster inhibition reaction time (ES ($g$) = 0.76; p = 0.039) and fewer errors across all tasks (color naming: $g$ = 0.8, p = 0.032; word reading: $g$ = 0.88, p = 0.019; inhibition: $g$ = 0.99, p = 0.009; switching: $g$ = 0.93, p = 0.012) following exercise intervention. Additionally, statistical analysis of the MMSE score showed a significant improvement ($g$ = 1.27; p = 0.001). In conclusion, our findings suggest that a short-term multicomponent exercise program improves selective tasks of executive function in postmenopausal women along with global cognitive function.

**Trial registration**

ISRCTN13086152

**Funding:** This research is funded by the Directorate General of Higher Education, Research, and Technology - Ministry of Education, Culture, Research, and Technology based on Decree Number 0536/E5/PG.02.00/2023 and contract agreement number 114/E5/PG.02.00.PL/2023; 1187/UN3.LPPM/PT.01.03/2023 The Ministry of Research, Technology and Higher Education of the Republic of Indonesia did not have any involvement in this study design, data collection or analysis the results, or writing the report.

**Competing interests:** The authors have declared that no competing interests exist.

## Introduction

In older women, declined cognitive function and mood changes are frequent complaints before, during, and after the menopausal transition. A recent review has identified that cognitive impairment such as lack of concentration, memory issues, planning difficulties, and brain fog, is commonly seen in 62%–67% of women during the menopausal transition [1]. This impairment continues to decline afterward [2,3]. Although the evidence of long-term changes in cognitive function during perimenopause and the progression remains unclear, postmenopausal women are more susceptible to a decline in cognitive function and neurodegenerative diseases. For example, women have a higher risk of Alzheimer's disease than men [4]. Furthermore, certain cognitive tests including the verbal memory test are gender sensitive [2].

The impact of menopause on the central nervous system is mostly caused by a rapid decline in the production of ovarian hormones, particularly estrogen. Estradiol, the most biologically active type of estrogen, has neuroprotective effects and elicits its effects through genomic and non-genomic signaling pathways to increase brain derived neurotrophic factor (BDNF) [5]. BNDF is responsible for the promotion of neuronal growth, maturation, and maintenance as well as synaptic plasticity and memory consolidation [5]. Moreover, estrogen plays a significant role to protect against oxidative stress-induced cell death in hippocampal neurons and interacts with the neurotransmitter system, thereby preventing neurodegeneration [6,7]. Furthermore, it interacts with insulin-like growth factor-1 (IGF-1), which stimulates neuronal cell survival and proliferation [7].

Although estrogen seems to contribute to cognitive decline during menopausal transition and afterward, the impact of estrogen replacement therapy on cognitive function in postmenopausal women remains inconclusive [8]. It is believed that the complex signaling pathways and molecular mechanisms of estrogen provide neuroprotective effects. The reduction in BDNF that accompanies the loss of estrogen may play a significant role in cognitive decline in postmenopausal women [5]. Therefore, strategies aimed to increase BDNF levels in postmenopausal women may be beneficial for maintaining cognitive function in postmenopausal women. Exercise has been proposed as one of the modalities for preventing a rapid decline in cognitive function in older adults. Several systematic reviews with meta-analysis have shown that exercise training in older adults, regardless of their cognitive status, has positive effects on cognitive function [9–12] in at least one cognitive domain. The mechanisms of exercise-induced improvement in cognitive function involve several pathways, including increased brain blood flow and growth of blood vessels, alteration of inflammatory cytokines (reduction in TNF-α and IL-6 concentrations), and increased levels of neurotrophic factors (BDNF) [13], independent of the estrogen-linked BDNF pathway [5]. Advanced research in humans, utilizing functional magnetic resonance imaging [14] reported that alterations in cognitive performance are mediated by changes in neural activation in different areas of the brain. Different types of physical activities are associated with the activation of distinct areas of the brain and cognitive functions [14].

The recent meta-analysis had already established the efficacy of exercise training for enhancing cognitive vitality in older adults; however, some moderator variables may interfere with the positive effect of exercise training on cognition [11,12,15]. Some studies [16,17] and meta-analysis [11,15] demonstrated that women gain more cognitive benefits from exercise training than men. Additionally, the benefits of exercise on cognition vary depending on the doses of exercise intervention (type, intensity, and duration), the characteristics of the participants at the beginning of the study (gender, general health, and fitness level), and the tasks used for measuring the aspects of cognition [11,15]. Executive function is a cognitive domain that is mostly affected by aging. However, this function less declined over time in women

compared to men [18]. Therefore, compared with men, the neural circuitry underlying this domain may remain more intact and may be more sensitive to targeted exercise intervention in women.

A recent study suggested that combining different types of exercises including cardiovascular, resistance, balance, and flexibility exercises can achieve greater gains in cognitive function [13]. Moreover, the World Health Organization strongly suggested older adults to engage in multicomponent exercises to enhance a wide range of physical functions, prevent falls, manage weight, increase bone health, and prevent osteoporosis [19]. However, evidence of the benefits that accrue with this type of exercise specifically for postmenopausal women is lacking. Furthermore, evidence of the short-term benefits of exercise program to cognitive function is limited. Therefore, this study aimed to investigate the short-term effects of a multicomponent exercise training program comprising aerobic (brisk walking), resistance, flexibility, and balance exercises on executive function together with global cognitive function in postmenopausal women. According to the latest fitness trend, aerobic and resistance training in the form of functional fitness as included in the exercise intervention in this study remain popular worldwide [20]. Several cognitive tasks have been used in studies evaluating the effects of exercise training on cognition, and the Stroop test and Mini-Mental State Examination (MMSE) were particularly selected as neuropsychological assessments to measure executive function and global cognitive function, respectively [21,22]. The Stroop test is sensitive to exercise training and consists of tasks that allow for the assessment of basic information processing and higher-order aspects of cognition associated with frontal lobes activity [21]. The MMSE is a global clinical, psychological, and neuropsychological indicator usually used for screening and evaluating the cognitive status of patients. It can comprehensively, accurately, and quickly reflect the intellectual quality and cognitive impairment of patients. We hypothesized that we could observe an improvement in executive function in addition to global cognitive function following a short-term multicomponent exercise intervention.

## Materials and methods

### Ethics statement and trial registration

This study has been approved by the Ethics Committee of the Faculty of Medicine, Airlangga University, Indonesia (reference number 44/EC/KEPK/FKUA/2023). After a thorough explanation of the testing protocol, the possible risks involved, and the right to terminate participation at any time, all participants provided informed consent before their enrolment into this study. Participant recruitment started on February 27, 2023, and ended on April 13, 2023. This study was registered in the International Standard Randomised Controlled Trial Number (registration number: ISRCTN3086152). This study was prospectively registered as the type intervention (exercise) allows us to register the trial as an ongoing/completed study after the first participant enrolment. The authors confirm that all ongoing and related trials for this intervention are registered.

### Participants' characteristic

The participants in this study were recruited from women's organizations that participated in the Family Welfare Empowerment Program in Surabaya, East Java, Indonesia. Those who met the initial eligibility requirements (women aged 50–80 years old who are at least 12 months postmenopausal and had a minimum educational level of junior high school) were invited to undergo the next stage of screening. Subsequently, the participants were screened to ensure that they did not have a history of stroke or any neurological problems, recent surgery, severe hearing or visual impairment, recent cardiovascular diseases (e.g., ischemic heart diseases and

heart failure), mental health problems, dementia, or moderate-to-severe cognitive impairment [defined as a score of <21 on the MMSE) [23]. Moreover, to ensure that they were not experiencing moderate-severe stress, psychological stress was assessed using the Depression Anxiety Stress Scale-21 (DASS-21).

### Study design

The Consolidated Standards of Reporting Trials (CONSORT) diagram outlining participants' flow from the first contact to the study completion is depicted in Fig 1. Participants who met the inclusion criteria were allocated to the exercise training group (EX, n = 15). Age-matched participants were recruited from similar women's organizations and allocated to the control group (CON, n = 15). Participants in the EX group underwent a 2-week exercise program that

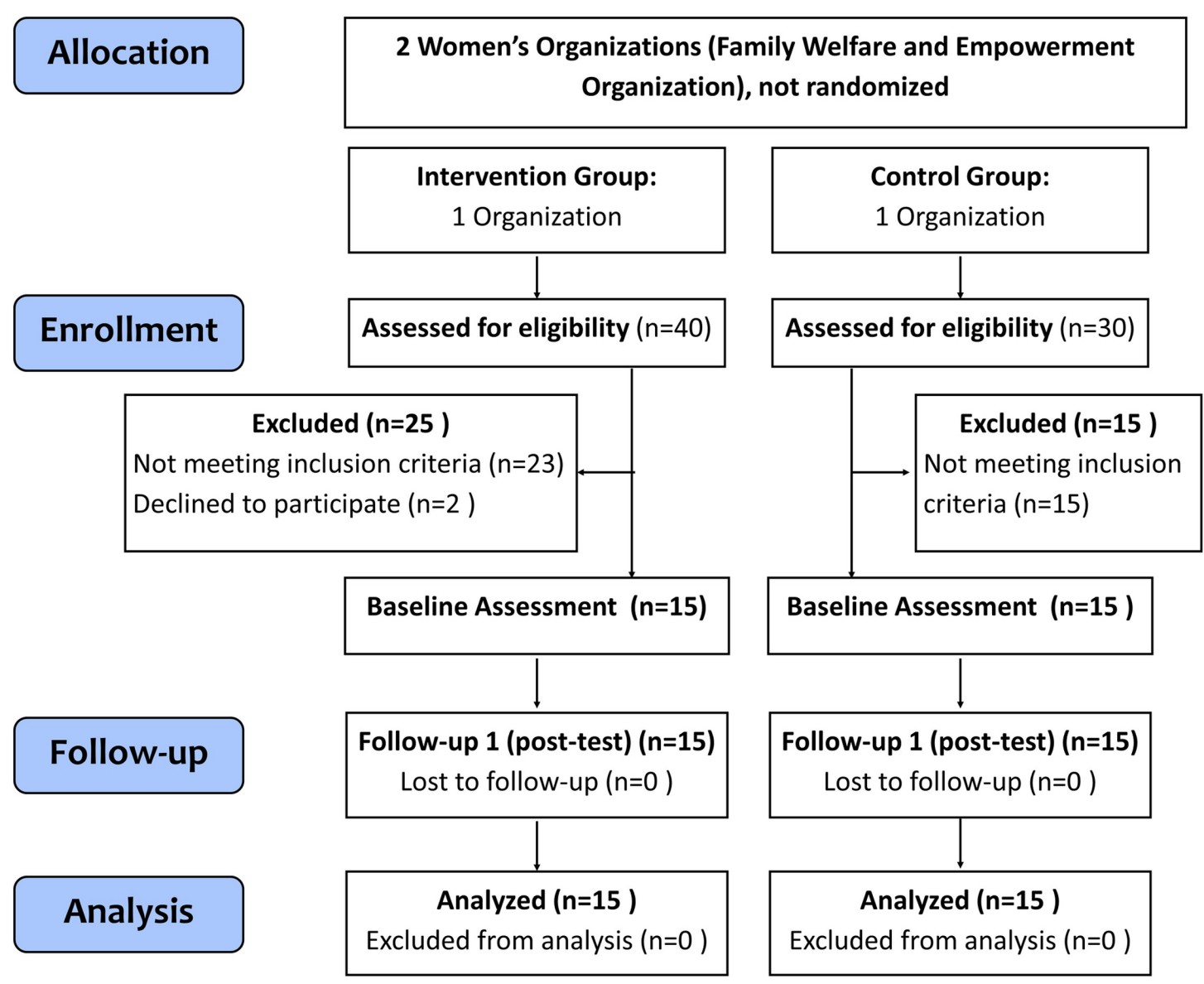

**Fig 1. CONSORT diagram of participants enrollment flow (modified for non-randomized study).**

included a combination of aerobic, strength, balance, and flexibility exercises. Conversely, those in the CON group were asked to continue with their usual activities. The short-duration exercise program was developed to be feasible in the community setting [24]. Additionally, cognitive improvement (verbal fluency test) was observed after a short program (14 days) of home-based exercise in combination with diet and relaxation in middle-aged participants (age, 35–69 years) [25]. The outcomes were assessed at baseline and 3 days following the final exercise session. At the initial meeting, all participants were asked to complete a form to collect their baseline information, including educational level, marital status, medical history, and medication. All the measurements were performed in a controlled-temperature room (25˚C) at the same time (06:00–08:00 AM). The data analyzer was blinded to the participant alloca-tion. Blinding of participants was not possible in this study. Additional information for study design and reporting may be found in S1 File. S2 File and S1 Checklist.

Following enrolment, a pretest was conducted 3 days before the first exercise session, and a posttest was conducted 3 days after the last exercise session. Before outcome measurements, all participants were asked to fast for 10 hours (last meal at 8 PM on the previous day) and refrain from smoking, consuming alcohol or carbonated drinks, and engaging in vigorous exercise. Upon participants' arrival, their blood glucose levels were measured, and they were allowed a light breakfast (2–3 pieces of biscuits, provided by the research team). Resting heart rate and blood pressure were measured two times using an automatic sphygmomanometer (OMRON Model HEM-7156; Omron Healthcare Co., Ltd, Japan). Global cognitive function and execu-tive functions were assessed using the MMSE and the Stroop test, respectively.

## Exercise training

Participants in the EX group attended exercise training five times/week for 2 weeks, every day on weekdays, in a community-based hall. Each exercise session was performed by groups. Each exercise session comprised a combination of aerobic (brisk walking), strength (using loop bands), balance, and flexibility exercises. In the first week, each session lasted for 40 min and progressively increased to 60 min. The overall intensity of exercise was moderate (RPE 5–6 of scale 10). The detail of exercise intervention protocol is provided in the S1 Table. The exercise was led by one instructor and two others who assisted and monitored the movements performed by the participants during the exercise. Participants' attendance was recorded in the logbook, and the adherence rate was calculated based on the number of completed training sessions.

## Outcomes measurement

**Executive function (Stroop test).** Executive function was assessed using a modified Stroop test [26] that consists of four different components and is adapted in Bahasa [27,28]. The intraclass correlation for reaction time and total error was 0.91 and 0.62 respectively [28]. Each component was printed on an A4 sheet of paper and consisted of practice (10 stimuli, five items per line) and test (50 stimuli, 10 items per line). The first condition was word read-ing, wherein the participants should read the words that were printed using black ink (e.g., red, green, blue, yellow). In the second condition (color naming), the participants should name the color of the rectangles. In the third condition (inhibition), the word was printed in different colors with their meaning. For example, the word "green" was printed using red ink. The participants should name the ink color of the printed words. Lastly, compared to the aforementioned tests, the fourth condition (inhibition/switching) was the most complex test. This test is similar to the third condition (inhibition); however, of 50 items printed color words; 20 were surrounded by a rectangle. In this condition, participants should read the

printed words instead of naming the color. Before each test, to ensure their understanding of the task, the participants were provided instructions and an opportunity to practice. For all conditions, participants should respond as fast as possible and point to which part they are currently on. They were allowed a maximum of four corrections for each test. All tasks were discontinued if the participants had (1) made four uncorrected errors or (2) reached the time limit for each task. The time limit for tasks 1 and 2 is 90 seconds, while the time limit for tasks 3 and 4 is 180 seconds. The completion testing time (reaction time, RT) was recorded in seconds. The total number of errors was determined by accumulating the corrected and uncorrected errors [29].

**Global cognitive health (MMSE).** To assess global cognitive function, including memory, attention, orientation, language, and visuospatial ability, we used the Indonesian version of MMSE. The MMSE score ranges from 0 to 30 points; in the Indonesian version, an MMSE score at or below 24 was classified as cognitive dysfunction or dementia. This cutoff point showed a sensitivity and specificity of 88% and 96%, respectively [30]. The Indonesian version MMSE had already undergone validation and reliability tests, with Cronbach alpha of 0.87 [31].

## Statistical analysis

The sample size calculation was calculated using G*power software version 3.1.9.7 based on the published data of Vaughan *et al.* (2014) [13] which reported the beneficial effects of a 16-week multimodal exercise program (including aerobic, strength, balance, flexibility, coordination, and agility exercises) on cognitive performance in older women. The study reported an effect size of 0.77 for the Stroop test (color naming). The minimum number of subjects required to establish significance is 14 per group would provide an 80% power ($\alpha = 0.05$) for F-test (ANOVA) repeated measures, within-between interaction.

All statistical analysis was computerized using software Jamovi 2.3.21.0 [32]. We performed the analysis of distribution using a visual inspection of Q-Q plot, skewness and kurtosis, and Shapiro Wilk test. The independent t-test was used to compare basic characteristics data of participants, whereas the outcome variables were assessed using linear mixed models. Linear mixed models were used to analyze differences in baseline and post-exercise cognitive performance. Time (pre vs. post) and group (control vs. exercise) were fixed factors, and participants were random factors to account for individual variations. We tested three covariance structures: compound symmetry, unstructured, and autoregressive, selecting compound symmetry based on the corrected Akaike Information Criterion (AIC). Based on that model, we investigated the interaction between time and group and the main effects separately (time: pre vs post and group: CON vs EX). For non-normal residuals, a bootstrap procedure with 10,000 resamples was used to obtain reliable standard errors and p-values. This mixed modeling can deal efficiently with missing data, unbalanced time points [33] and robust with the violations in distributional assumptions [34]. This means that the missing data could be included in the analyses, without imputation. This method also compensates for selective dropout, on the condition that dropout is related to variables included in the models. However, there were no missing data in this study. The analyses were carried out by the per-protocol approach. Only those who attended a minimum of eight exercise sessions (80%) were included in the post-test. Estimated means, standard deviations, 95% confidence intervals, and the estimation of the interaction between time and treatment in the model are presented. Bonferroni test was used for post-hoc test, with a significance level of 95%. Effect sizes of the improvements observed according to Hedge's *g* statistic were computed as the absolute change of the outcome measure to its pooled SD in the group assessed before and after training. Effect size were interpreted as

trivial (<0.2), small (0.2 to 0.49), moderate (0.50 to 0.79), and large ($\geq$0.80) [35]. Graphs were made using PRISM 9.5.1 (Graph Pad Software, La Jolla, CA, USA).

## Results

### Participant characteristics

Of the 15 participants enrolled in the EX group, the adherence to the exercise program was 98.7%. The baseline characteristics of the participants are presented in Table 1. No significant differences in age, body mass index, resting heart rate, and blood pressure were observed (all p > 0.05) (Table 1). Two participants in the EX group were diagnosed with type-2 diabetes mellitus and were treated with oral antidiabetic medication. Three participants (two and one participant in the CON group and EX group, respectively) were diagnosed with hypertension and were treated with an oral antihypertensive medication. Other information on participant's characteristics is shown in Table 1.

**Table 1. Participant's baseline characteristic.**

| | Control (n = 15) | Exercise (n = 15) | P-Value (Independent T-test) |
|---|---|---|---|
| Age, year | 58.27 ± 5.44 | 60.27 ± 4.63 | 0.116 |
| Height, cm | 154.00 ± 3.43 | 152.90 ± 6.50 | 0.557 |
| Body Weight, kg | 59.20 ± 7.36 | 59.53 ± 9.95 | 0.919 |
| BMI, kg/m$^2$ | 24.90 ± 2.49 | 25.53 ± 4.28 | 0.629 |
| Resting heart rate, beats/min | 79.53 ± 8.46 | 81.90 ± 9.29 | 0.472 |
| Blood pressure | | | |
| Systolic blood pressure, mmHg | 127.40 ± 18.63 | 132.30 ± 14.40 | 0.427 |
| Diastolic blood pressure, mmHg | 78.87 ± 10.15 | 83.63 ± 7.92 | 0.163 |
| Mean arterial pressure, mmHg | 95.01 ± 12.22 | 99.89 ± 9.47 | 0.232 |
| FPG mg/dL | 117.4 ± 32.9 | 128.70 ± 57.09 | 0.515 |
| Education | | | 0.424 |
| Junior high school | 0 (0%) | 3 (20%) | |
| Senior high School | 7 (46.6%) | 10 (66.6%) | |
| Diploma | 1 (6.6%) | 2 (13.3%) | |
| Bachelor | 7 (46.6%) | 0 (0%) | |
| Medical History | | | 0.484 |
| Diabetes mellitus | 0 (0%) | 2 (13.3%) | |
| Hypertension | 2 (13.3%) | 1 (6.6%) | |
| History of cardiovascular diseases | 0 (0%) | 0 (0%) | |
| History of cerebrovascular diseases | 0 (0%) | 0 (0%) | |
| Current or ex-smoker | 0 (0%) | 0 (0%) | |
| Medication | | | 0.899 |
| Oral antidiabetic | 0 (0%) | 2 (13.3%) | |
| Antihypertensive | 2 (13.3%) | 1 (6.6%) | |
| Mental Health (DASS-21 Score) | | | |
| Depression | 1.33 ± 1.39 | 1.87 ± 3.22 | 0.564 |
| Anxiety | 2.46 ± 1.95 | 3.40 ± 3.15 | 0.341 |
| Stress | 3.66 ± 2.89 | 3.13 ± 3.31 | 0.646 |

BMI, Body Mass Index; DASS-21, Depression Anxiety Stress Scale-21; FPG, Fasting Plasma Glucose. All data are presented as mean ± standard deviation (SDs).

*Significantly different from the control group at p < 0.05.

## Effects of exercise on cognitive test

**Stroop test performance.** The results of the Stroop test on completion time performance (reaction time) and total errors before and after exercise are presented in Table 2 and Figs 2 and 3. In the word reading performance, we detected non-significant differences from the baseline of the EX group versus the CON group (p = 0.081; *g* = 0.64; 95% CI: -0.09, 1.37),). However, a significant large reduction of total error was evident in this task when compared to the baseline in the EX group versus the CON group (p = 0.019; *g* = 0.88; 95% CI: 0.12, 1.62).

Regarding color naming reaction time, we did not find a significant difference from the baseline of the EX group versus the CON group (p = 0.916; *g* = 0.04; 95% CI: -0.67, 0.76). In the total errors, we detected a large significant reduction when comparing differences from the baseline of the EX group versus the CON group (p = 0.032; *g* = 0.8; 95% CI: 0.05, 1.54).

In the inhibition task, we observed a significant large reduction when comparing differences from the baseline of the EX group versus CON group of the reaction time (p = 0.039; *g* = 0.76; 95% CI: 0.01, 1.49) and total errors (p = 0.009; *g* = 0.93; 95% CI: 0.17, 1.68).

In the inhibition/switching task, we did not observe a significant reduction when comparing differences from the baseline of the EX group versus the CON group of the reaction time (p = 0.168; *g* = 0.51; 95% CI: -0.23, 1.23). Additionally, we detected a significant large reduction when comparing differences from the baseline of the EX group versus the CON group of the reaction time (p = 0.012; *g* = 0.93; 95% CI: 0.17, 1.68).

**Table 2. Stroop test and MMSE assessment at baseline (pre) and week 2 (post).**

| | Control (n = 15) | | | | Exercise (n = 15) | | | | Time*Group | | | ES *g* |
|---|---|---|---|---|---|---|---|---|---|---|---|---|
| | Pre | | Post | | Pre | | Post | | mean (SE) | 95% CI | p | |
| | mean (SE) | 95% CI | mean (SE) | 95% CI | mean (SE) | 95% CI | mean (SE) | 95% CI | | | | |
| **Stroop test - Reaction time, second** | | | | | | | | | | | | |
| Word Reading[a] | 25.41 (0.96) | 24.07, 27.66 | 25.03 (0.9) | 23.94, 27.47 | 30.12 (1.92) | 26.44, 32.51 | 26.1 (1.47) | 22.89, 28.88 | -3.57 (1.97) | -8.39, 1.21 | 0.081 | 0.64 |
| Color naming[a] | 34.09 (1.06) | 33.24, 36.34 | 32.51 (1.08) | 31.64, 34.61 | 37.73 (2.18) | 33.46, 40.88 | 35.91 (1.52) | 31.96, 38.63 | -0.22 (2.1) | -5.71, 4.62 | 0.916 | 0.04 |
| Inhibition[a] | 61.87 (2.25) | 55.72, 80.16 | 70.45 (8.68) | 65.46, 94.79 | 87.09 (13.13) | 65.46, 94.79 | 68.06 (5.34) | 49.96, 76.26 | -27.08 (12.5) | -56.25, 2.54 | 0.039 | 0.76 |
| Inhibition/ Switching[a] | 104.99 (12.55) | 80.98, 138.31 | 91.19 (12.05) | 69.36, 127.31 | 137.41 (14.27) | 103.56, 155.5 | 93.42 (12.14) | 68.78, 113.78 | -29.69 (20.99) | -78.7, 23.56 | 0.169 | 0.51 |
| **Stroop test - Total Error** | | | | | | | | | | | | |
| Word Reading[a] | 0.07 (0.07) | -0.37, 0.31 | 0.07 (0.07) | -0.36, 0.33 | 0.73 (0.28) | 0.48, 1.16 | 0.07 (0.07) | -0.18, 0.51 | -0.66 (0.27) * | -1.21, -0.12 | 0.020 | 0.87 |
| Color naming[a] | 0.6 (0.16) | 0.27, 0.96 | 0.2 (0.14) | -0.04, 0.64 | 1.47 (0.39) | 0.75, 2.1 | 0.07 (0.07) | -0.4, 0.63 | -0.99 (0.44) * | -1.98, -0.01 | 0.032 | 0.8 |
| Inhibition | 1.8 (0.41) | 0.74, 3.25 | 1.87 (0.57) | 0.91, 3.43 | 3.8 (0.76) | 2.23, 4.75 | 1.47 (0.36) | 0.02, 2.54 | -2.38 (0.84) * | -4.1, -0.65 | 0.009 | 0.99 |
| Inhibition/ Switching[a] | 3.8 (0.82) | 2.61, 5.26 | 3.67 (0.75) | 2.73, 5.43 | 5.93 (0.72) | 3.96, 6.89 | 3.2 (0.76) | 1.86, 4.52 | -2.55 (0.95) * | -4.91, -0.02 | 0.012 | 0.93 |
| | | | | | | | | | | | | |
| MMSE score (0–30) | 25.33 (0.61) | 24.18–26.56 | 26.47 (0.39) | 25.22–27.6 | 24.2 (0.64) | 23.07, 25.47 | 28.47 (0.39) | 27.23, 29.61 | 3.11 (0.87)* | 1.33, 4.89 | 0.001 | 1.27 |

Values are from linear mixed effects models adjusted for age, years of education, and DASS-21 score.

[a] Bootstrap for non-normal distribution of the residuals.

ES: Effect size.

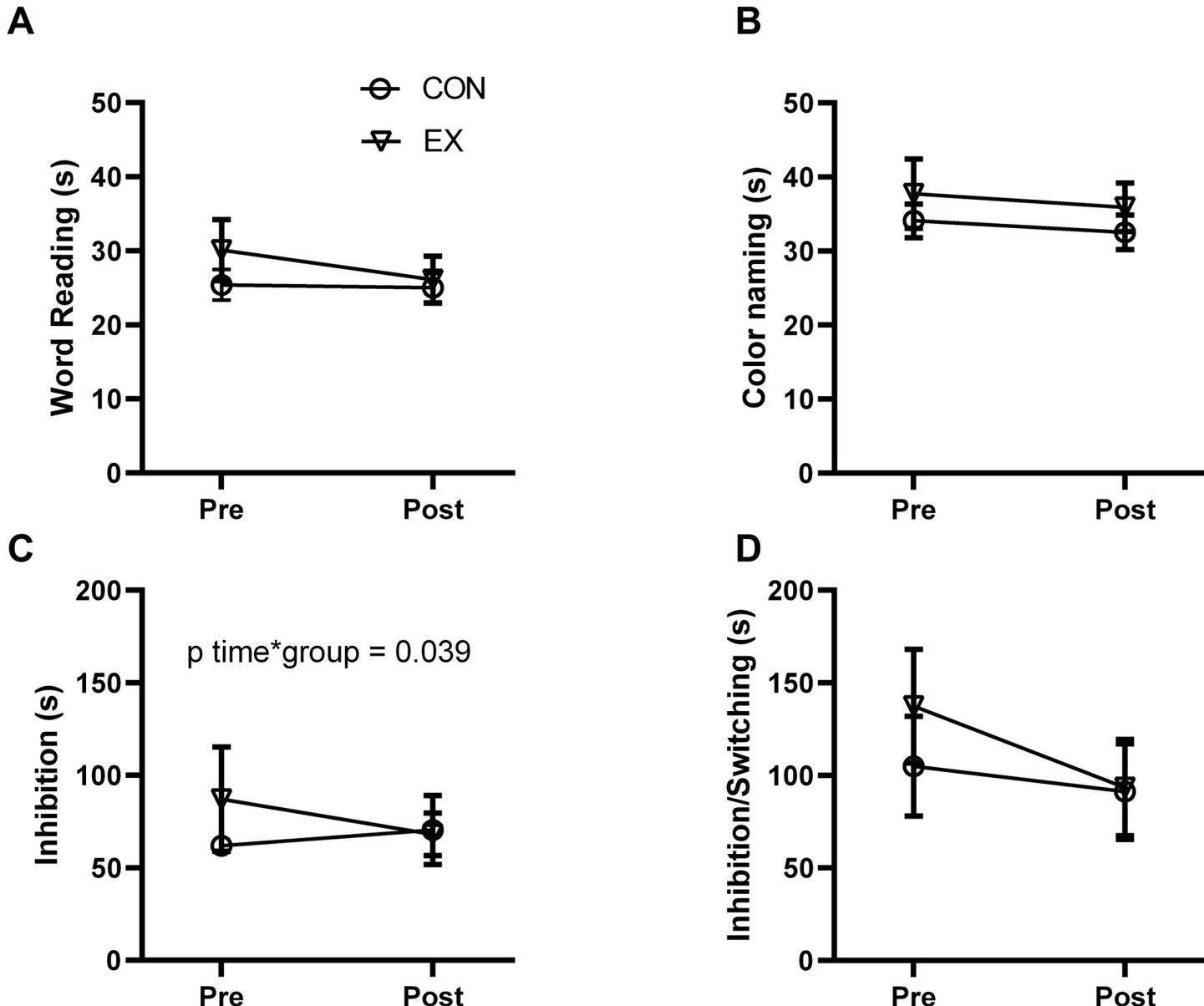

**Fig 2. Effects of exercise intervention on reaction time of the Stroop test performance in postmenopausal women.** Reaction time to complete the word reading task (top, left panel–A), color naming task (top, right panel–B), inhibition task (bottom, left panel–C), and inhibition/switching task (bottom, right panel–D) before (pre) and after (post) 2-weeks. Data is presented as means ± 95% confidence intervals (CI). Effects are derived from linear mixed models, adjusted for age, years of education, and DASS-21 score.

### The effect of exercise on global cognitive function (Mini-Mental State Examinations - MMSE)

The analysis of MMSE is presented in Table 2 and Fig 4. We observed a significant large improvement in MMSE score when comparing differences from the baseline of the EX group versus the CON group (p = 0.001, $g$ = 1.27; 95% CI: 0.47, 2.05).

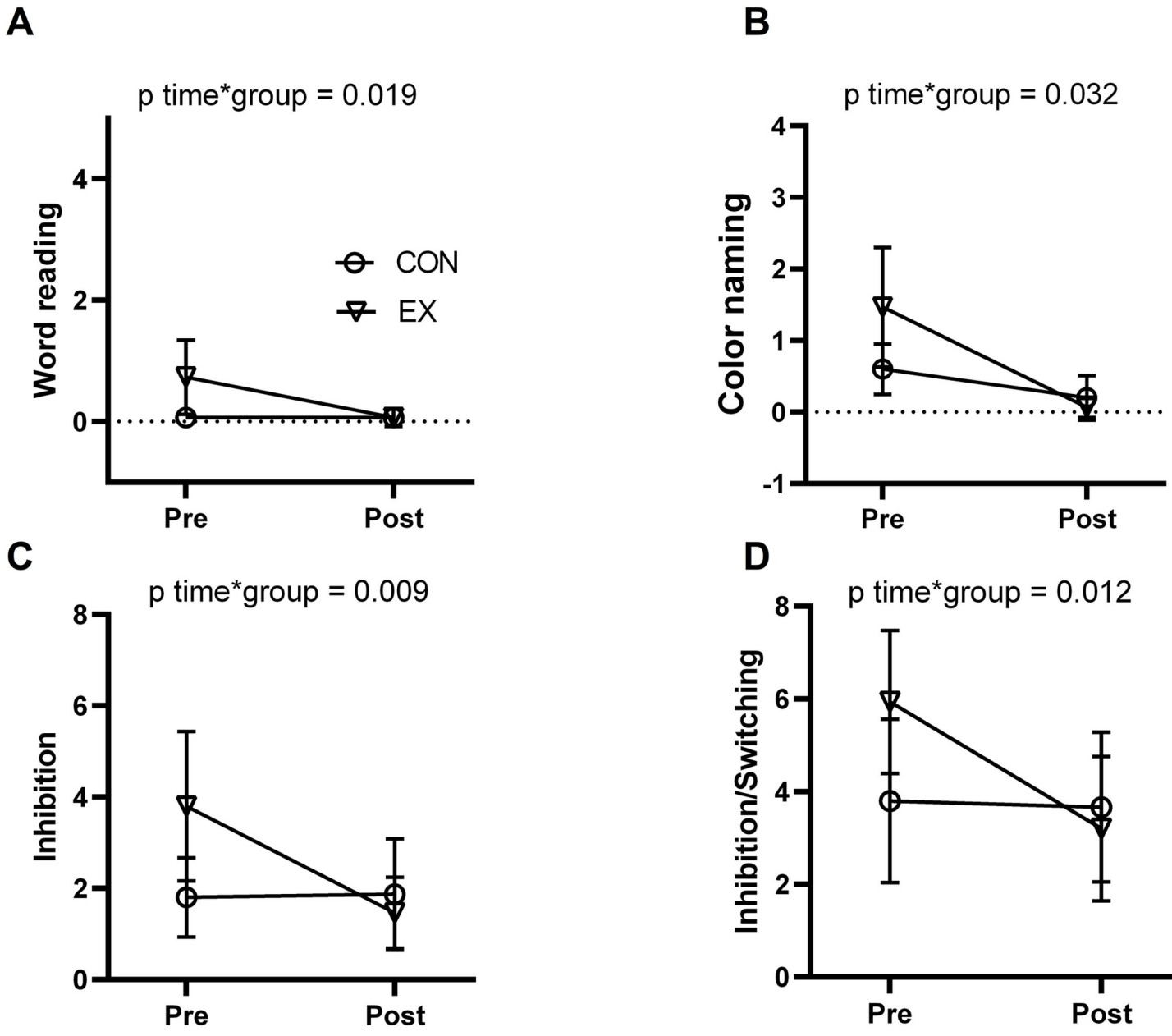

**Fig 3. Effects of exercise intervention on total errors of the Stroop test in postmenopausal women.** Total errors during the completion of the word reading task (top, left panel–A), color naming task (top, right panel–B), inhibition task (bottom, left panel–C), and inhibition/switching task (bottom, right panel–D) before (pre) and after (post) 2-weeks. Data is presented as means ± 95% confidence intervals (CI). Effects are derived from linear mixed models, adjusted for age, years of education, and DASS-21 score.

## Discussion

This study evaluated the effects of multicomponent exercise training programs on cognitive function in postmenopausal women. The main findings of this study were 1) the improvement of inhibition task performance of the Stroop test following a short-term multicomponent exercise training program. We also noted a positive trend towards improvement of the reaction time of other components of the Stroop task (color naming, word reading, and switching) although did not reach statistical significance. Moreover, we observed the improvement in

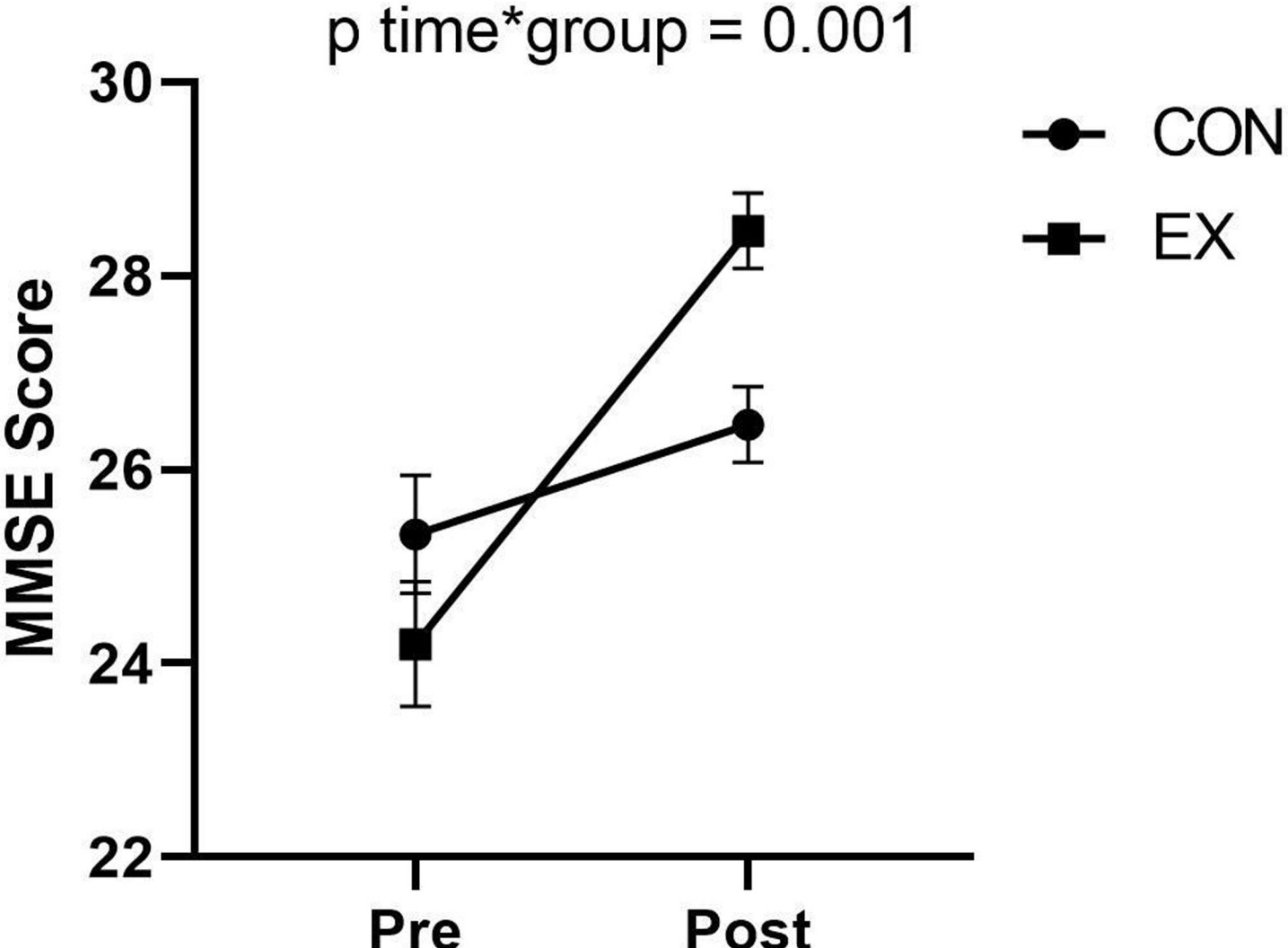

**Fig 4. Effects of exercise intervention on MMSE in postmenopausal women.** The graph shows MMSE scores before (pre) and after (post) 2-weeks of exercise training in the exercise group and the control group. Data is presented as means ± SE. Effects are derived from linear mixed models, adjusted for age, years of education and DASS-21 score.

accuracy in performing the Stroop test in all tasks as shown in the reduction of total errors; 2) the improvement of global cognitive function using the MMSE.

### Effects of multicomponent exercise training programs on cognitive function

Our results in Stroop test performance were consistent with a previous study that investigated the effects of a 16-week multicomponent exercise (aerobic-strength-motor fitness) in older women. The results demonstrated a moderate-to-large effect size of exercise to improve the reaction time of all tasks of the Stroop test, including color naming, word reading, and inhibition tasks [13]. In contrast, another study demonstrated no significant effect in all components of the Stroop tasks following a 6-month multicomponent exercise program in older women [36]. However, this study used a different analysis of the Stroop test by the total number of correct words in 45 seconds instead of the reaction time and total error [36]. Most studies that

investigated the effect of exercise employed aerobic exercises. One study evaluated executive functions after a 12-week of aerobic training in older adults (>70% of women participants) and noted differences only in the switching condition. Similarly, significant improvement in total error was observed only in the switching component [29]. Another study of aquarobics program in older adults (>70% women participants) showed improvement in the Stroop test performance after 21 weeks but not in a shorter-term intervention (10 weeks) [37]. Although generally longer periods of exercise intervention provide more benefit in cognitive function, however, the meta-analysis study showed that shorter duration (4–12 weeks) leads to better cognitive improvement than medium duration (12–24 weeks). Additionally, this meta-analysis showed that combined aerobic and resistance exercise demonstrated a larger effect compared to aerobic alone [11].

A possible explanation of the discrepancy between our results with those of other studies is the differences in the modality and doses of exercises. The effect of physical exercise on cognitive function has been proven to be task-specific and motor activity-specific [38]. Previous study hypothesized that a higher cognitive reserve resulting from physical exercise is because of a neural compensation mechanism that permits complex activities [39]. Complex tasks such as inhibition and inhibition/switching in the Stroop test require multiple executive processes. Other study suggested that tasks requiring more control and effortful processing including the Stroop test, should be more sensitive to fitness differences among older adults than tasks that can be processed automatically [40].

In this study, we estimated a moderate effect of exercise on the speed improvement of the inhibition task of the Stroop test and a larger effect on the accuracy of all tasks in the postmenopausal women population. Our positive findings on executive function suggest that exercise has a beneficial effect on controlling functions critical for processes involved in cognitive learning. These functions include information processing speed, attention control, resistance to interference, dual-tasking, and cognitive flexibility [13]. Recent systematic review research has suggested a significant cognitive decline in executive processes during progressive neuropsychological follow-ups, leading to a potential manifestation of mild cognitive impairment (MCI) or dementia [41]. From a clinical perspective, a 15-year prospective study on individuals with Alzheimer's disease revealed that lower cognitive performance at the executive level began to show more pronounced signs 2–3 years before receiving a clinical diagnosis [42]. Thus, monitoring the performance of executive function tasks and developing interventions to maintain or improve these cognitive functions is necessary to address the epidemic of dementia and other cognitive disorders. However, the generalization of our findings should be approached with caution as our study focused on post-menopausal older women, a group that may be psychologically vulnerable [4] and may potentially experience more benefit in executive processes from exercise than men [15]. In addition, the executive function as the primary outcome of this study is one of the cognitive domains that commonly benefited positively from exercise interventions. Similar interventions might have different effects on other cognitive domains such as verbal fluency.

The second aim of our study was to investigate the global cognitive function using the MMSE. Regarding the MMSE score, our results were in line with the previous study although this study showed little effect on the MMSE score improvement. This may be because the participants in this group have already high MMSE scores at baseline (mean score > 27) [36]. This test is believed to have a ceiling effect because the highest score is 30. Moreover, our findings are consistent with the results of a meta-analysis study that showed most exercise interventions improve global cognitive function in older individuals [9].

Our study did not investigate the mechanism of cognitive function improvement as a result of exercise intervention. However, several studies have shown that physical exercise improves

brain functioning through both structural and functional changes. Studies have reported that exercise training can lead to an increase in gray and white matter volume in the prefrontal cortex and hippocampal regions [43]. The prefrontal area is important to process the executive function and this cognitive task has the most pronounced effect because of exercise training [11]. Our findings showed that short-term multicomponent exercise training improves both executive and general cognitive functions. This is probably because the multicomponent exercise training program not only improves physical fitness but also enhances motor learning skills, including balance and flexibility. A cross-sectional study suggested that physical and motor fitness were differentially related to cognitive functioning, while physical fitness was mainly related to the executive control process, motor fitness showed a significant association with both the executive control and perceptual speed tasks. Further brain imaging analysis revealed that physical fitness counteracts the effects of aging in the prefrontal and temporal areas. Conversely, motor activity is more likely to enhance visuomotor coordination and visuospatial integration in the parietal area of the brain [14]. Therefore, maintaining an "enriched exercise program" in older age is necessary. Additionally, the most effective modality to improve health-related outcomes in specific populations such as adults with overweight/ obese is combining different types of exercises during the same session [44]. Strategy for exercise intervention to improve cognitive performance or ameliorate cognitive decline in older women is related to the modality, dose of the exercise, and social support. Combining at least one different type of exercise with aerobic training has more significant effects on cognition, such as a combination of aerobic and resistance [11,12] or multicomponent exercise [15] as performed in this present study. Regarding the dose of exercise, moderate bouts of exercise duration (> 30–60 min) [11,12] and moderate intensity [9,12] on as many days of the week as possible [12] were suggested. Exercise types including cognitively engaging exercises have the most pronounced effect on ameliorating cognitive decline in older adults with mild cognitive impairment [9]. Furthermore, exercise in a group of similar age will increase adherence to exercise in older individuals [45]. To the best of our knowledge, our study is the first to report the beneficial effects of short-term multicomponent exercise on the executive function of postmenopausal older women.

## Strengths and limitations

The strength of the present study includes high adherence levels to the exercise program. Herein, we reported the high compliance level of the participants to the exercise program (adherence, >90%). However, monitoring of daily physical activity in the control or exercise group was not conducted using objective tools such as activity-tracking tools. Instead, an initial consent letter was utilized to discourage participants from engaging in additional physical activities. Moreover, encouragement was provided through text messages to each participant in both groups to help them maintain their daily physical activities. Furthermore, our study also reported that a 2-week multicomponent exercise program improved cognitive performance in postmenopausal women and demonstrated an average moderate to large effect size.

Despite the positive effects observed in the present study, there were some limitations. First, the selection of participants into groups was not randomized, which may have influenced the heterogeneity of the baseline values of the outcomes. To mitigate the negative effects of the nonrandomized design, we recruited age-matched participants as controls and conducted an initial screening of cognitive function using the MMSE score. Second, our results may be confounded by baseline activity levels, health status, and medication used. None of the participants met the criteria for active living as outlined in the physical activity guidelines established by the American College of Sports and Medicine for healthy older adults, i.e., five times a week,

30 minutes of moderate-intensity physical activity. In terms of health status, we included participants with risk factors for cognitive decline and dementia, such as type 2 diabetes mellitus (T2DM) and hypertension. All of them were taking medications for their respective conditions. Previous study demonstrated that adherence to antihypertensive medication was associated with a change in global cognitive performance. However, there was no indication of a connection between adherence to oral hypoglycemic agents and cognitive health [46]. Although both diseases have been associated with an increased risk of cognitive decline and dementia, a meta-analysis revealed that exercise training in the T2DM population was beneficial for cognitive performance [47], whilst the available randomized controlled trials (RCTs) regarding the effects of physical exercise (PE) on cognitive performance in individuals with hypertension are limited [48]. The studies mentioned above support that there was no strong evidence that these participants experienced reduced cognitive benefits from the exercise intervention. Therefore, further study may be warranted to address the potential differential effects of multicomponent exercises in individuals with type 2 diabetes mellitus (T2DM) and/or hypertension. Third, our findings are specific to postmenopausal older women and may not be applicable to other populations, including male participants.

## Conclusions

This study suggests that short-term multicomponent exercise can improve certain aspects of cognitive function in postmenopausal women, particularly in the accuracy of performing the Stroop test and global cognitive function. Further studies that include male participants are needed to generalize the results to the overall aging population.

## Supporting information

**S1 Checklist. TREND statement checklist for reporting nonrandomized controlled trials.** (PDF)

**S1 Table. Exercise intervention protocol.** (PDF)

**S1 File. Research protocol in Bahasa Indonesia.** (PDF)

**S2 File. Research protocol in English.** (PDF)

## Acknowledgments

We extend our gratitude to the participants and Ms. Uswatun Hasanah and Fath., M.D. for their assistance with this study.

## Author Contributions

**Conceptualization:** Dani Rahmat Ramadhana, Rizki Prayuda Putra, Sulistiawati Sulistiawati, Dewi Ratna Sari, Purwo Sri Rejeki, Lilik Herawati, Raden Argarini.

**Data curation:** Dani Rahmat Ramadhana, Raden Argarini.

**Formal analysis:** Dani Rahmat Ramadhana, Sulistiawati Sulistiawati, Raden Argarini.

**Funding acquisition:** Dani Rahmat Ramadhana, Sulistiawati Sulistiawati, Raden Argarini.

**Investigation:** Dani Rahmat Ramadhana, Rizki Prayuda Putra, Michelle Abigail Sibarani.

**Methodology:** Dani Rahmat Ramadhana, Rizki Prayuda Putra, Michelle Abigail Sibarani, Sulistiawati Sulistiawati, Purwo Sri Rejeki, Lilik Herawati, Raden Argarini.

**Resources:** Dani Rahmat Ramadhana, Rizki Prayuda Putra, Michelle Abigail Sibarani, Raden Argarini.

**Supervision:** Sulistiawati Sulistiawati, Dewi Ratna Sari, Raden Argarini.

**Validation:** Raden Argarini.

**Visualization:** Dani Rahmat Ramadhana, Raden Argarini.

**Writing – original draft:** Dani Rahmat Ramadhana, Raden Argarini.

**Writing – review & editing:** Dani Rahmat Ramadhana, Rizki Prayuda Putra, Michelle Abigail Sibarani, Sulistiawati Sulistiawati, Dewi Ratna Sari, Purwo Sri Rejeki, Lilik Herawati, Raden Argarini.

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
