## [Decision Letter · Decision Letter 0]

26 Dec 2023

PONE-D-23-28550Short-term combined exercise training improves cognitive function in post-menopausal womenPLOS ONE

Dear Dr. Argarini,

Thank you for submitting your manuscript to PLOS ONE. After careful consideration, we feel that it has merit but does not fully meet PLOS ONE’s publication criteria as it currently stands. Therefore, we invite you to submit a revised version of the manuscript that addresses the points raised during the review process.

We look forward to receiving your revised manuscript.

Kind regards,

Thiago P. Fernandes, PhD

Academic Editor

PLOS ONE

Journal Requirements:

2. The in-house editorial staff feels that your study meets the World Health Organization definition of a clinical trial because it is a prospective study in which participants were assigned to an exercise intervention to investigate the effects on cognitive function. Please change your manuscript’s article type to ‘Clinical Trial’ when you resubmit your manuscript.

In addition, please provide the following important pieces of information/documentation.

1) Please note that you must upload a completed CONSORT flowchart as figure 1 of your manuscript and the

CONSORT checklist as a supporting information file. Blank copies of these documents and information

regarding CONSORT can be found via the following link: http://www.consort-statement.org/. If your

clinical trial uses a non-randomized design, you may wish to submit a TREND checklist

(http://www.cdc.gov/trendstatement), in place of the CONSORT checklist; a flowchart is still required.”

2) Please upload a copy of your trial study protocol as a supporting information file. By the study protocol, we

mean the complete and detailed plan for the conduct and analysis of the trial that the ethics committee

approved before the trial began. Please send this in the original language. If this is in a language other than

English, please also provide a translation. Please detail any deviations from this study protocol in the

Methods section of your manuscript. Your study protocol will be made available to the editors and

reviewers, and will be published as supporting information with your manuscript if accepted for publication.

(If you do not agree to this, we will not be able to publish your manuscript). If you have formally published a

study protocol for your trial in a journal then you should cite this in your manuscript, but you still need to

send us the original document.

3) PLOS ONE requires that all clinical trials are registered in an appropriate registry (the WHO list of approved registries is at

http://www.who.int/ictrp/network/primary/en/index.html and more information on trial registration is at

http://www.icmje.org/about-icmje/faqs/clinical-trials-registration/).

Please state the name of the registry and the registration number (e.g. ISRCTN or ClinicalTrials.gov) in the

submission data and on the title page of your manuscript.

If you have not yet registered your trial in an appropriate registry, we now require you to do so and will need confirmation of the trial registry number. Please include in the Methods section of your paper your reasons for not registering this study before enrolment of participants started. Please confirm that all related trials are registered by stating:

“The authors confirm that all ongoing and related trials for this drug/intervention are registered”.

Please see http://journals.plos.org/plosone/s/submission-guidelines#loc-clinical-trials for our policies on

clinical trials.

Additional Editor Comments:

Thank you for submitting your valuable work.

The reviews, which are insightful and interesting, pointed to some unexplained aspects. The authors will notice the reviewers found merits in your study, but also raised important concerns. Nevertheless, authors also will notice that this may sound lengthy, but the main intention is to work more closely with the suggestions (while also ensuring the high standard of scientific communication).

I was reluctant to provide inputs in this round, but upon re-reading the comments, I think the authors address them in due time.

1. The Title would benefit from increased specificity. For instance, specifying the cognitive functions and types of exercises would enhance clarity. As I understand that this can be the authors' perspective to make it more 'punchy', I don't think there's this need to change a lot. Consider: a. the Title effectively convey the study's main idea and objectives? and b. From a broader audience perspective outside the field, is there any options to improve understanding?;

2. The Abstract needs improvement by providing exercise descriptions, addressing strengths and limitations, and refining sentences for readability. Consider placing mean age after "50-80 years" and use an abbreviation other than the one provided for exercise to avoid confusion;

- Ensure decimal consistency;

- For the aged groups, maybe the MoCA, instead of the MMSE, would be a good option. As this might cause misinterpretation by researchers and readers outside the field, consider tempering language re. "global function." Avoid capitalising "pre" and "post" and refine terms for clarity;

- Emphasise effect sizes over p-values;

3. Provide additional context on how menopause influences cognitive function. The text lacks background and rationale, crucial for understanding authors' exercise choices and their neural links;

4. Please provide more details on the sample, especially demographics. Provide a detailed overview of eligibility, recruitment, and workflow;

5. Enhance transparency by detailing data recording and adherence to best practices making data available;

6. Consider including more details on tests, such as reliability coefficients, Cronbach's alpha. Please consider the use of illustration to explain the procedures for the tests. Also present the extracted variables;

7. Please refine the stats section with information on effect sizes, confidence intervals, and rationale for tests used for analyses. Why use a risky post-hoc? Any corrections to? Consider that your model can be inflated and your analyses would be better interpreted if you could simplify the models;

8. I'd highly suggest checking again your parameters, skewness, kurtosis and the chosen tests. Seems like the model without considering interaction within- is worrying. I genuinely think that LMM is more appropriate. Also, please consider that PRISM not always provides the reliable details - you can consider other handy software that are quick and won't take that much time, like Jamovi or JAPS;

9. Clearly explain how missing data were handled if observed;

10. Please, if possible, utilise dispersion data in graphs, configuring options for boxes and scatter, for example;

11. Consider that Discussion needs a little bit working, tightening up to your findings and what previous studies observed;

12. I think the authors could consider to refine Discussion presenting the strengths and limitations;

13. Carefully check the refs. list for adherence to Journal's standards;

Overall, the study is, indeed, interesting and conducted in the proper way, but the lack of specifications, details and the stats presentation are the main hindrances of this version. Apart from it, theoretical contributions could be given for readers and researchers, so they can understand the generalisation and the relevance of your study - even if they are outside the field.

Hope the authors find all (or most) of the suggestions useful and hope this helps somehow.

Remember to adhere to the standards of the journal and make, if possible, your data available.

The authors are highly encouraged to enlist a native English speaker to make some edits.

Wishing you success with the study.

if you find yourself approaching the due date, feel free to request a bit more time.

Please do not be concerned about time constraints; simply demonstrate the same level of dedication you exhibited during the study.

Reviewers' comments:

Reviewer's Responses to Questions

**Comments to the Author**

1. Is the manuscript technically sound, and do the data support the conclusions?

Reviewer #1: Partly

Reviewer #2: Yes

Reviewer #3: Yes

Reviewer #4: Yes

Reviewer #5: Yes

2. Has the statistical analysis been performed appropriately and rigorously? 

Reviewer #1: Yes

Reviewer #2: Yes

Reviewer #3: Yes

Reviewer #4: Yes

Reviewer #5: Yes

3. Have the authors made all data underlying the findings in their manuscript fully available?

Reviewer #1: No

Reviewer #2: Yes

Reviewer #3: Yes

Reviewer #4: Yes

Reviewer #5: Yes

4. Is the manuscript presented in an intelligible fashion and written in standard English?

Reviewer #1: No

Reviewer #2: Yes

Reviewer #3: Yes

Reviewer #4: Yes

Reviewer #5: Yes

5. Review Comments to the Author

Reviewer #1: In this research authors investigated the effects of a multi-component exercise program on cognitive function of post-menopausal women. The study deals with an interesting topic. However it has some important limitations: very short training program; lack of randomization; lack of a proper control group

Introduction:

Authors should specify which is the first study endpoint (according to what is stated in the statistical paragraph it should be changes (post versus pre exercise) in Stroop test.

Method

The choise of a short exercise training program in the present study (only two weeks) need to be justified and supported with previous researches.

In my opinion some important aspects of the training program (such as intensity of different exercises) are missing. Authors should provide these details in the method section

It is not clear to me why the authors gave up on randomization.

Discussion

In my opinion this paragraph is more like a second introduction. Therefore it should be re-organized. Results obtained in his research should be compared with previous similar researches and differences should be underlined.

Reviewer #2: General comments

The authors have clearly stated that the purpose of the study was to evaluate investigate the effects of short-term combined exercise training programs on cognitive function in post-

menopausal women. The paper is well-written, easy to follow and adds merit to the vital role of exercise training in the cognitive function. Given this approach, this work can enhance future attempts in similar research area. However, I have highlighted a few suggestions and concerns in my specific comments section (below) that need to be addressed before considering whether this work should be published or not. In summary, this is an innovative and work that might be considered for publication after revising the initial manuscript as needed.

Specific comments

ABSTRACT

- Replace mean values ± SD with Δ% and and effect sizes per outcome measure in results.

INTRODUCTION

- Add a sentence about the beneficial role of combined training in health among adults according to the latest guidelines by the World Health Organization (1).

- Add a statement about the popularity of both aerobic and resistance training in the health and fitness industry worldwide according to the latest report published by the American College of Sports Medicine (2).

Suggested References:

1. https://pubmed.ncbi.nlm.nih.gov/33239350/

2. https://journals.lww.com/acsm-healthfitness/pages/articleviewer.aspx?year=2022&issue=01000&article=00007&type=Fulltext

RESULTS

- 95% confidence intervals and effect sizes should be added in both the text and tables for all outcome measures.

DISCUSSION

- Discuss the results observed for combined training with respect to various populations (3) as a critical piece of overall health improvements.

Suggested reference:

3. https://pubmed.ncbi.nlm.nih.gov/35477256/

- Strengths and limitations should be presented in a separate paragraph at the end of the discussion section. A heading is also needed.

- In conclusions, you should underline the main findings and suggest future research attempts in this area while highlighting potential practical implications. This paragraph needs a significant improvement.

Reviewer #3: The research design of this study is relatively reasonable, the experimental control is relatively appropriate, and the data statistics are relatively rigorous. The result data can support the conclusion.The logical thinking and language expression of the article are relatively clear. However, the author did not seem to indicate whether the experiment concealed the experimental purpose from the participants (blind design). Will this cause some bias in the research results? In addition, the author does not seem to clearly indicate why "Stroop Test" and "Mini Mental State Examination" were chosen to evaluate the cognitive function of the participants. If a more detailed explanation of these two assessment methods (the use of previous scholars) is added in the introduction part, will it be better ?

Reviewer #4: It is an interesting topic, Let me understand the effect of short-term comprehensive training on the cognitive function of postmenopausal women. But I think some parts of the article need further revision to ensure the quality of the article.

*Summary*

The overall structure of the article is still a little confused, should focus on modification.

Point 1: The manuscript should be checked by a native speaker before publication.

Point 2: There are too many old references that are more than 10 years old; please replace all possible references, with articles within the last 5 years.

*Introduction*

Point 3:Page1 Line54-56. “Furthermore,.....sensitive”.This paragraph says that women are more likely to get Alzheimer's disease, but how is it directly relevant to this article, or to clarify its relationship to cognitive function in postmenopausal women.

Point 4:The second paragraph focuses on the fact that exercise can improve cognitive function in the elderly. But there was no mention of whether exercise improved cognitive function in postmenopausal women. It is because there is no similar literature research, or not written in the article, I suggest adding.

Point 5:Page4 Line75-83. This paragraph again talks about the relationship between motor and cognitive function, similar to the content of the second paragraph, it is recommended to merge.

Point 6: The research background is not very detailed, and it is not clear why such research is done, because of the lack of previous studies, or what other reasons.

*Materials and methods*

Point 7:Page4 Line23-24. “ They were also screened to ensure that they are not having specific diseases as mentioned earlier in the inclusion and exclusion criteria”. This sentence is repeated with the inclusion criteria of Participants characteristic, so it is recommended to delete it

Point 8:Page4 Line28. “The psychological stress was also assessed using the Depression Anxiety Stress Scale-21 (DASS-21) to ensure that they were not experiencing moderate-severe stress”.It is suggested to put this sentence in the previous paragraph.

Point 9:Page6 Line118-120.Whether the experimental group and the control group were assigned randomly, and by what specific way.If it is not a randomized controlled trial, please indicate what kind of study it is.

Point 10:Page6 Line124-125. “All the measurements were performed in a controlled-temperature room (25°C) at the same time (06:00 – 08:00 AM)”.It is suggested to put this sentence in the previous paragraph

Point 11:It is recommended to include a picture to show how the subjects were assigned, whether there were drop-outs, and the number of people analyzed at the end.

Point 12:Page7 Line150.Change the Cognitive test to Outcome measurements.

*Resutls*

Point13:“P＜0.05”should be write as“p＜0.05”.

*Discussion*

The discussion section is well written

Point 14: In the discussion section, some specific strategies for exercise intervention in cognitive function of postmenopausal women should be added.

Reviewer #5: A two-arm controlled study was conducted which aimed to investigate the effects of short-term combined exercise training programs on cognitive function in post-menopausal women. After a two-week exercise program, the study showed an improvement in global cognitive function of the exercise group.

Minor revisions:

1- Line 183: Indicate the statistical testing method which achieves 80% power.

2- Indicate if the distribution of the data was checked for normality prior to applying t-tests or ANOVA.

3- Perform tests of interactions prior to testing main effects. If the interaction effect is significant, provide an interpretation of the results, but do not test main effects because the tests for main effects are uninteresting in light of significant interactions. If interaction effects are non-significant, drop the interaction effects from the model and test the main effects. Determining which results to present when testing interactions is often a multi-step process.

4- Indicate the date range subjects participated in the study.

6. PLOS authors have the option to publish the peer review history of their article (what does this mean?). If published, this will include your full peer review and any attached files.

Reviewer #1: **Yes: **Giuseppe Caminiti

Reviewer #2: No

Reviewer #3: No

Reviewer #4: No

Reviewer #5: No

---

## [Author Response · Author response to Decision Letter 0]

13 Mar 2024

Dear Dr. Fernades,

Thank you for reviewing our manuscript. and the opportunity to revise our manuscript. We have made changes accordingly. We have submitted the main documents, including a rebuttal letter, track changes, and a clean version of the manuscript, along with supplementary materials such as the protocol and trial registration, which are required for the revision.

We apologize that we have made a mistake in our financial disclosure. We have made revisions to the financial disclosure as detailed below

"This research is funded by the Directorate General of Higher Education, Research, and Technology - Ministry of Education, Culture, Research, and Technology based on Decree Number 0536/E5/PG.02.00/2023 and contract agreement Number 114/E5/PG.02.00.PL/2023; 1187/UN3.LPPM/PT.01.03/2023"

I look forward to hearing from you in this regard.

Kind Regards,

Raden Argarini

---

## [Decision Letter · Decision Letter 1]

5 Apr 2024

PONE-D-23-28550R1Short-term multicomponent exercise training improves executive function in postmenopausal womenPLOS ONE

Dear Dr. Argarini,

Thank you for submitting your manuscript to PLOS ONE. After careful consideration, we feel that it has merit but does not fully meet PLOS ONE’s publication criteria as it currently stands. Therefore, we invite you to submit a revised version of the manuscript that addresses the points raised during the review process.

We look forward to receiving your revised manuscript.

Kind regards,

Thiago P. Fernandes, PhD

Academic Editor

PLOS ONE

Journal Requirements:

**Additional Editor Comments:**

Thank you for submitting your valuable work.

Along with the remaining and important concerns to be addressed, please consider adapting a few parts:

1. It would be beneficial for the study to include a detailed explanation of how test scoring was conducted. Specifically, clarifying whether errors were calculated and describing the methods used to measure and analyse the outcomes would enhance understanding and replicability;

2. The adaptation of the Stroop test for Bahasa use is an interesting aspect of this study. However, lacks details on the standardisation procedures and the validation process of the adapted version. Providing this information is crucial for ensuring the test's reliability and validity, thereby strengthening the study's findings;

3. While it's commendable that intraclass correlation coefficients for reaction time and total error were included, the absence of information on inter-rater reliability for scoring the Stroop test is noted. Inter-rater reliability is important for confirming that scoring consistency is maintained across different evaluators, and its inclusion would solidify confidence in the scoring methodology;

4. The choise to utilise LMMs for assessing outcome variables is not explicitly justified. Elaborating on the reasons for choosing this modeling approach over others, such as ANOVA or regression models, would provide insight into its advantages for your analysis;

5. The ms briefly mentions employing a bootstrap procedure but didn't explain the specifics of its implementation or its influence on the reliability of SE and p-values. A more detailed description would aid in evaluating the statistical inference's robustness, enhancing the credibility of the findings;

6. Providing comprehensive details on scoring methodologies, standardisation, and validation processes will not only clarify the study's approach but also enable replication and verification by other researchers;

7. The study commendably reports high adherence to the exercise program, yet the lack of objective measures to monitor physical activity levels in the control group may undermine the validity of the intervention-control comparisons. Incorporating objective activity tracking, such as wearable devices, could strengthen the study's findings by ensuring accurate activity monitoring across all participants;

8. While the focus on postmenopausal women provides valuable insights, the specific tailoring of the intervention to this group limits the broader applicability of the findings. Elaborate on how other could explore similar interventions in diverse demographic groups to enhance generalisation;

9. The study's methodology could be enhanced by more thoroughly accounting for potential confounding factors such as education, socioeconomic status, and health conditions. Adjusting for these variables in the analysis would provide a clearer understanding of the intervention's effect;

10. While the study reports on effect sizes, expanding the discussion to include the clinical relevance of these effects would offer deeper insights into their practical implications for cognitive function in postmenopausal women;

11. While acknowledging the study's non-randomised design is important, a deeper exploration of other potential confounders, including baseline activity levels and medication use, would strengthen the study's conclusions by clarifying influences on the observed outcomes;

Reviewers' comments:

Reviewer's Responses to Questions

**Comments to the Author**

1. If the authors have adequately addressed your comments raised in a previous round of review and you feel that this manuscript is now acceptable for publication, you may indicate that here to bypass the “Comments to the Author” section, enter your conflict of interest statement in the “Confidential to Editor” section, and submit your "Accept" recommendation.

Reviewer #2: All comments have been addressed

Reviewer #4: (No Response)

Reviewer #5: (No Response)

2. Is the manuscript technically sound, and do the data support the conclusions?

Reviewer #2: Yes

Reviewer #4: (No Response)

Reviewer #5: Yes

3. Has the statistical analysis been performed appropriately and rigorously? 

Reviewer #2: Yes

Reviewer #4: (No Response)

Reviewer #5: Yes

4. Have the authors made all data underlying the findings in their manuscript fully available?

Reviewer #2: Yes

Reviewer #4: (No Response)

Reviewer #5: Yes

5. Is the manuscript presented in an intelligible fashion and written in standard English?

Reviewer #2: Yes

Reviewer #4: (No Response)

Reviewer #5: Yes

6. Review Comments to the Author

Reviewer #2: I have no additional comments. The revised manuscript is ready for publication. Congratulations to the authors on their work.

Reviewer #4: (No Response)

Reviewer #5: Minor revisions:

1- Line 219: Indicate the statistical testing method which achieve 80% power. For example statistical testing methods are t-tests or chi-square tests, etc.

2- Indicate the underlying covariance structure used in the linear mixed model and the criteria for selecting it.

3- Indicate the number of iterations used in the bootstrap procedure.

7. PLOS authors have the option to publish the peer review history of their article (what does this mean?). If published, this will include your full peer review and any attached files.

Reviewer #2: No

Reviewer #4: No

Reviewer #5: No

---

## [Author Response · Author response to Decision Letter 1]

20 May 2024

20 May 2024

Dr. Emily Chenette 

Editor in Chief

PLoS ONE

Dear Dr. Emily Chenette,

Thank you for reviewing our manuscript and the opportunity to revise our manuscript. We have made changes accordingly.

We have submitted our second revision documents including a rebuttal letter, track changes, and a clean version of the manuscript.

Should you need other information, please do not hesitate to contact us.

I look forward to hearing from you in this regard.

Kind Regards,

Argarini

---

## [Editor Report · Decision Letter 2]

23 May 2024

PONE-D-23-28550R2Short-term multicomponent exercise training improves executive function in postmenopausal womenPLOS ONE

Dear Dr. Argarini,

Thank you for submitting your manuscript to PLOS ONE. After careful consideration, we feel that it has merit but does not fully meet PLOS ONE’s publication criteria as it currently stands. Therefore, we invite you to submit a revised version of the manuscript that addresses the points raised during the review process.

Thank you for your valuable submission. I think the authors addressed all remaining concerns, and we can proceed with the study. Nevertheless, I request the authors to check the file I am sending because some values and calculations are not aligned, incorrect, or need to be recalculated. Apart from this and the need to check grammar throughout the file and verify the refs, the manuscript was improved. I do not think this will take a lot of time, so I will await these updates.Once these concerns are addressed, we can streamline the phase. If you weren't able to see the annotations, please email. Wishing you success with the study

We look forward to receiving your revised manuscript.

Kind regards,

Thiago P. Fernandes, PhD

Academic Editor

PLOS ONE

Journal Requirements:

Additional Editor Comments:

Please check the file.

---

## [Author Response · Author response to Decision Letter 2]

7 Jul 2024

Dear Editor Team,

We have revised our manuscript according to your comments in the previous version.

Thank you

---

## [Editor Report · Decision Letter 3]

12 Jul 2024

Short-term multicomponent exercise training improves executive function in postmenopausal women

PONE-D-23-28550R3

Dear Dr. Argarini,

We’re pleased to inform you that your manuscript has been judged scientifically suitable for publication and will be formally accepted for publication once it meets all outstanding technical requirements.

Kind regards,

Thiago P. Fernandes, PhD

Academic Editor

PLOS ONE
---

## [Editor Report · Acceptance letter]

18 Jul 2024

PONE-D-23-28550R3 

PLOS ONE

Dear Dr. Argarini, 

I'm pleased to inform you that your manuscript has been deemed suitable for publication in PLOS ONE. Congratulations! Your manuscript is now being handed over to our production team.

Kind regards, 

on behalf of

Dr. Thiago P. Fernandes 

Academic Editor

PLOS ONE